

# Nutritional status and its associated factors among commercial female sex workers in Hawassa city, south Ethiopia

Zegeye Gelan[1], Zelalem Tafese[1], Eskinder Yilma[2] and Mahbuba Kawser[3]

[1] School of Nutrition, Food Science and Technology, College of Agriculture, Hawassa University, Hawassa, Sidama, Ethiopia
[2] School of Public Health, College of Health Science and Medicine, Wolaita Sodo University, Wolaita Sodo, SNNPR, Ethiopia
[3] Institute of Nutrition and Food Science, University of Dhaka, Dhaka, Bangladesh

Corresponding author
Zegeye Gelan, zegeyegt1@gmail.com

## ABSTRACT

**Background.** Commercial female sex workers (CFSWs) regularly or occasionally trade sex for money. Sex work is widespread, mainly in urban areas of Ethiopia. The study on the nutritional status of CFSWs is unavailable in Ethiopia, and data are also scanty at the global level. This study aims to assess nutritional status and associated factors among CFSWs in Hawassa city, Ethiopia.

**Methods.** This cross-sectional study used facility-based mixed data collection methods (qualitative and quantitative). The study was conducted in three key population clinics in Hawassa city. A total of 297 CFSWs were randomly selected for the 'quantitative survey,' and twelve ($n = 12$) purposively selected participants were involved in the 'qualitative study.' Body mass index/BMI ($Kg/m^2$) was used in assessing the nutritional status of CFSWs. Statistical software packages were used for analyzing both quantitative and qualitative data. Significant variables ($P < 0.05$) in bivariate analysis (Chi-square test) were incorporated in the multivariable analyses. Multinomial logistic regression (MNLR) was employed where dependable variable like 'normal BMI' ($18.5–24.9 \, kg/m^2$) was set as a reference category and simultaneously compared with 'underweight' ($<18.5$ BMI) and 'overweight/obesity' ($\geq 25$ BMI) categories. Thus, two models, namely the underweight model (model-1: Underweight versus normal BMI) and the overweight/obesity model (model-2: overweight/obesity versus normal), were generated.

**Results.** The prevalence of underweight and overweight/obesity among CFSWs in Hawassa city were respectively 14.1% and 16.8%. Living alone (Adjusted odds ratio/AOR = 0.18), chewed Khat regularly (AOR = 0.23), used drugs regularly (AOR = 10.57), used drugs in exchange of sex (AOR = 4.97), and HIV positive status (AOR = 21.64) were significantly ($P < 0.05$) associated with underweight (model-1). In the overweight/obesity model-2, having jobs other than sex work (AOR = 0.11), higher daily mean income (AOR = 3.02), being hotel/home-based CFSWs (AOR = 12.35), and presence of any chronic illness (AOR = 5.15) were significant ($P < 0.05$) predictors of overweight/obesity. From the 'qualitative part' of this study, it was also revealed that 'lack of food and money' was the main influencing factor among CFSWs to enter into the sex business.

**Conclusions.** Commercial female sex workers in this study faced a double burden of malnutrition. Multiple factors influenced their nutritional status. Substance abuse and

HIV-positivity are the most significant predictors of being underweight and having higher income, being hotel/home-based CFSWs, and suffering from any chronic illness are associated with overweight/obesity. Government and other partners should be essential in providing comprehensive programs focusing on sexual, reproductive health, and nutrition education. Steps should be taken to improve their socioeconomic status and strengthen those good initiatives at key population clinics and other health facilities.

## INTRODUCTION

Female sex workers (FSWs) are defined in the Ethiopian context as females who regularly or occasionally trade sex for money in drinking establishments, nightclubs, local drinking houses, "Khat" and "Shisha" houses. They also worked on or off street settlements, around military and refugee camps, construction sites, trade routes, red-light districts, and at their homes. It was also reported that 11.71% of female sex workers in Ethiopia are mostly young, between the ages of 15–24 (*Federal HIV/AIDS Prevention and Control Office, HAPCO, 2015*; *Ethiopia Public Health Institute, EPHI, 2014*), and girls from Ethiopia's impoverished rural areas are exploited in domestic servitude and commercial sex within the country (*US State Department, 2017*). Recent estimates show about 200,000 commercial female sex workers (CFSWs) in Ethiopia (*US President's Emergency Plan for AIDS Relief, PEPFAR, 2018*). Moreover, in a study conducted in eighty-nine Ethiopian towns, 85,294 CFSWs were counted there, and one-third were non-self-identified (*Girma et al., 2014*).

Sex workers are among the key populations most affected by HIV and STIs, especially in African countries where HIV prevalence was higher even in the general population than the other nations (*Ethiopia Public Health Institute, 2022*; *Hladik et al., 2017*; *United Nations Programme on HIV/AIDS, UNAIDS, 2021*; *World Health Organization, 2018*). The HIV prevalence among sex workers in sub-Saharan Africa in 2021 was 12%, while it was 30.6% for the east and southern sub-region of Africa among sex workers (*United Nations Programme on HIV/AIDS, UNAIDS, 2021*). The World Health Organization (WHO) reported that key populations, clients, and sexual partners accounted for 25% of new HIV infections in Africa's east and southern sub-region (*World Health Organization, 2018*). The overall estimated HIV prevalence was 33% in Kampala, the capital city of Uganda, while it was 44% among female sex workers of $\geq 25$ years (*Hladik et al., 2017*). In 2022, the estimated HIV prevalence among Ethiopian adults (15+ years) was 0.9%, while it was 3.2% among the adult population of Hawassa city (*Ethiopia Public Health Institute, 2022*). However, a previous study (2013) showed much higher (23%) HIV prevalence among Ethiopian female sex workers (*Ethiopia Public Health Association/EPHA, 2013*).

Studies reported that married, very young, and illiterate women serving as on-street sex workers are dominant in the Asian sex industry. Most accepted the sex-selling profession because of deceptions and served $\geq 3$ clients per night with minimal payment (*Mondal,*

*Rahman & Hossain, 2010*; *Mahejabin, Nahar & Parveen, 2014*). Despite widespread coercion in the sex industry, many sex workers have to work in the sex business to support themselves and their families (*Hengartner et al., 2015*). Adolescents forced into the commercial sexual exploitation of women and children suffer multiple forms of abuse, which may be life-threatening (*Bagley et al., 2017*). Female sex workers suffered from different types of severe health complications but spent very few amounts of money on health purposes also reported (*Mondal, Rahman & Hossain, 2010*). Legal issues, stigma, discrimination, and violence pose barriers to HIV services for sex workers (*United Nations Programme on HIV/AIDS, UNAIDS, 2012*).

A study reported that sex workers could not take food properly due to their addiction, floating habit, and lack of money (*Huq et al., 2013*). Globally, research on sex workers' nutritional well-being was not studied widely, although thousands of studies have been undertaken on 'sex work' and the health problems of 'sex workers' (*United Nations Programme on HIV/AIDS, UNAIDS, 2014*). A recent study conducted in a south Asian country showed that the prevalence of underweight and overweight/obesity among CFSWs in Dhaka was 22.8% and 16.3% (*Kawser et al., 2020*). However, the nutrition-related burden of sex works and sex workers has not yet been at the forefront of research in Ethiopia, even though in African countries, despite having a higher burden of HIV/AIDS and sexually transmitted infections (STIs) (*Ethiopia Public Health Institute, 2022*).

According to the Ethiopian Demographic and Health Survey, the prevalence of underweight among reproductive-aged women in Ethiopia was 17.6% (*Kassie, Abate & Kassaw, 2020a*; *Kassie et al., 2020b*). Moreover, the prevalence of malnutrition among people living with HIV and AIDS (PLWHA) was 60% (*Nigusso & Mavhandu-Mudzusi, 2021*). The prevalence of undernutrition and overweight/obesity was 29.3% and 2.4% among PLWHA in Bahir Dar city, Ethiopia (*Hussien et al., 2021*). The study reported (*Nigusso & Mavhandu-Mudzusi, 2021*) that female gender, urban residence, income below 53.2 US dollars (USD), poor asset possession, duration of less than one year on anti-retroviral therapy/ART, and recurrent episodes of opportunistic infections were strongly associated with malnutrition among PLWHA. A statistically significant association was also reported between body mass index (BMI, kg/m$^2$) and hypertension (*Tesfa & Demeke, 2021*).

Furthermore, the dietary diversity of sex workers had not been assessed before in Ethiopia, even in greater Africa. Studies conducted in Africa (*Anyanwu et al., 2021*; *Gitagia et al., 2019*) reported that the proportion of dietary diversity scores for women (DDS-W) was low (consumption of <5 food groups) and not meeting the minimum DDS threshold for women (≥5 food groups). United Nations (UN) has specially acknowledged emphasizing the health needs and rights of women belonging to "vulnerable groups," including those in prostitution (*United Nations, 1999*). Moreover, it was recommended that state parties take measures to protect persons working in the sex industry against all forms of violence, coercion, and discrimination (*CECR, 2016*).

Considering the poor socioeconomic status, less access to a hygienic environment, risky sexual behaviors, drug and alcohol taking, and the infection-prone status of commercial female sex workers, and, more importantly, given the importance of the higher prevalence

of HIV among CFSWs of Hawassa city and no current research on nutrition-related burden among them, this study perceived to assess the nutritional status of commercial sex workers and to identify its associated factors in Hawassa city, south Ethiopia.

## MATERIALS & METHODS

### Study sites, design, and study participants

The study was conducted in Hawassa city, Ethiopia, to assess the nutritional status of commercial female sex workers and to determine the factors associated with underweight and overweight/obesity. 'Hawassa city' is in the Sidama region, Ethiopia, and is located on the shores of lake 'Hawassa' in the Great Rift Valley, 273 km south of Addis Ababa, the capital city of Ethiopia. It lies on the Trans-African highway for Cairo-Cape Town and has a latitude and longitude of 7°32N and 38°282 E coordinates and an elevation of 1,708 m above sea level (*Wikipedia, 2022*). Based on the projection of the 2007 census conducted by the 'Central Statistical Agency of Ethiopia,' the city has an estimated total population of 442,900 in 2021 (*Government of Ethiopia, 2020*).

A facility-based cross-sectional study using quantitative and qualitative methods was conducted from December 2021 to January 2022. Systematic 'random sampling' methods for the quantitative and purposive sampling techniques for the qualitative study were employed to select the study participants. There are two government-owned (Adare General Hospital and Millennium Health Center) and one non-government organization (NGO) owned (*e.g.*, Family Guidance Association of Ethiopia/FGAE Clinic) commercial female sex workers clinics which serve as 'key population clinics' in the city. Latter mentioned NGO-owned CFSW clinic is confidential and provides comprehensive health services for only female sex workers and their partners. A total of 297 CFSWs from three female sex workers' clinics in Hawassa city were included in the 'quantitative study', with a response rate of 99.6%. One sex worker refused to participate in the study. Moreover, twelve ($n =$ 12) CFSWs were selected for in-depth and key informant interviews (qualitative part) to support the quantitative data.

### Sample size determination

The sample size was determined according to the following single population proportion formula:

$$N = (z_{1-a/2})^2 p\,(1-p)/d^2$$

where p = Prevalence of under-nutrition ($p = 22.8\%$) (*Kawser et al., 2020*), Z = 1.96, d = degree of precision (d = 0.05 or 5%), the significance level of study = 95%, Therefore, N = 271.

Adding a non-response rate of 10% to the calculated sample size, the final sample size required for this study was 298.

### Data collection instrument and methods
#### Quantitative data

In the quantitative or main part of the study, data collection was conducted using pre-tested, structured, and interviewer-administered questionnaires. Based on the findings of

pre-testing, the questionnaire was modified before the final interviews. The questionnaire contains socio-demographic characteristics, personal and sexual behaviors, and health practices among sex workers. A systematic random sampling technique was employed to select the study participants (Fig. S1 & Table S1). Greater than 18 years older CFSWs who gave informed consent were included in the study. Female sex workers who were pregnant and lactating mothers, seriously ill, intoxicated, and unable to respond to the data collection time were excluded from the study. Five trained healthcare providers conducted face-to-face interviews to collect the quantitative data. The enumerators were professionally experienced (mainly nurses and health officers working in sex worker's clinics) and fluent in the Amharic language.

### Qualitative data

In-depth interviews (IDI) and key informant interviews (KII) were used in the qualitative study to complement the quantitative survey. Twelve ($n = 12$) CFSWs were purposively selected for the qualitative study. In-depth interviews and KII guides were followed to conduct and record the interviews. The interviews were recorded by using a Sony digital voice recorder. Individuals who participated in the quantitative data collection and the pre-testing session (during the training of the interviewers) were excluded from the qualitative study.

### BMI assessment

After completing the interview, body weight was measured to the nearest 0.1 kg, wearing no shoes with light clothing, on a portable weighing scale. Standing height was measured with a wall-mounted scale to the nearest 0.1 cm, with the head in the Frankfurt horizontal plane, while standing straight on a horizontal surface with heels together, the shoulders relaxed, arms at the sides, and without shoes. Height and weight were used to calculate Body Mass Index (formerly known as Quetelet index), BMI = weight (kg)/height ($m^2$). The classifications of body mass indices, like <18.5 kg/$m^2$, 18.5–24.9 kg/$m^2$, and ≥25 kg/$m^2$, represented underweight, normal, and overweight/obese (*World Health Organization, WHO, 2004*).

### Dietary intake assessment

Dietary diversity score for women (DDS-W) was estimated among CFSWs using a quantitative 24-h recall method. Minimum dietary diversity for women (MDD-W) is a dichotomous indicator that depicts whether or not women of 15–49 years of age have consumed at least five ($n = 05$) out of ten ($n = 10$) defined food groups on the previous day or night. The proportion of women (15–49 years) who reaches this minimum level can be used as a proxy indicator for higher micronutrient adequacy, one crucial dimension of diet quality. The ten food groups are: (1) grains, white roots and tubers, and plantains; (2) pulses (beans, peas, and lentils); (3) nuts and seeds; (4) dairy; (5) meat, poultry, and fish; (6) eggs; (7) dark green leafy vegetables; (8) other vitamin A-rich fruits and vegetables; (9) other vegetables and (10) other fruits (*Food and Agriculture Organization, FAO, 2016*).

## Ethics

Ethical clearance to conduct the study was obtained from the Institutional Review Board (IRB), School of Graduate Studies of Hawassa University, (ref No: IRB/021/14). Written permission to carry out the study was also obtained from the respective administrative bodies and study facilities. Both verbal and written consent was collected from study participants. Before the survey, all questionnaires and consent forms were translated into the Amharic language.

## Quality control

A three-day training was provided to all five interviewers (professional health care providers), and the questionnaire was pre-tested in another drop-in-center of Hawassa city. Collected data were cross-checked for content validity and completeness. The principal investigator conducted in-depth and key informant interview/KII sessions during the quantitative data collection period.

## Statistical analyses

A statistical software package (SPSS version 20.0; IBM, Chicago, IL, USA) was employed to analyze the quantitative data. Before performing analysis, the normality test of the BMI was done by the Kolmogorov–Smirnov (K–S) goodness of fit test. Entered data were coded, transformed (where necessary), and cleaned for further analysis. The results were presented in the form of tables or figures using summary statistics (*e.g.*, frequencies (n), percentage (%), mean, and standard deviation/SD) to describe the study population. Body mass index/BMI ($Kg/m^2$) with three categories (underweight, normal, and overweight/obese) was used in assessing the nutritional status (dependable/outcome variable) of the CFSWs. 'Multicollinearity test' was performed through checking variable inflation factors (VIF <1). In bivariate analysis (Table S2), cross-classification analysis ($X^2$-test) was used to examine possible associations of different independent variables or predictors (*e.g.*, socio-demographic, health, and sexual behaviors) with the dependable variable BMI ($kg/m^2$). *P*-value of <0.05 was set as significant. Significant variables ($P < 0.05$) in bivariate analysis (Chi-square test) were incorporated in the multivariable analyses. Multinomial logistic regression (MNLR) was used to predict the independent factors associated with the prevalence of nutritional status (BMI $kg/m^2$). The multinomial logistic regression is a simple extension of binary logistic regression that allows for more than two categories of the dependent/outcome variable. In the MNLR analyses, 'normal BMI' (18.5–24.9 $kg/m^2$) was set as a 'reference category' and simultaneously compared with 'underweight' (<18.5 BMI) and 'overweight/obesity' ($\geq$25 BMI) categories through estimating two logit equations (log of the odds, is the coefficient provided by a logistic regression). Exponentiation was used to convert logits to the odds ratios, thus function "exponential or exp (logit)/(1+exp (logit)" can be used to convert logits to the probabilities. The 'forward stepwise elimination' process was used for 'variable selection'. The final model was created based on variables significantly associated at the final step of variable selection. The 'model selection' was based mainly on the Akaike Information Criterion (AIC) (or Bayesian Information Criterion/BIC). Thus, two models were regenerated: model-1, namely the underweight model (Underweight

*versus* normal BMI), and model-2, or the overweight/obesity model (Overweight/obesity *versus* normal). The findings were presented as odds ratios ((Exponential (exp) value of beta-coefficient or exp (B)) with 95% confidence intervals. The Log likelihood ratio ($L^2$), Pearson's goodness of fit, Nagelkerke Pseudo-$R^2$ values, and classification table for BMI ($kg/m^2$) were considered for fitting regression models. In general, non-significant *P*-value (Pearson's goodness of fit $P > 0.05$), and lower AIC (or BIC) values in relation to a smaller log likelihood ratio or $L^2$ suggests a 'well model fit' (*Anderson & Rutkowski, 2008*; *Westfall & Henning, 2013*).

Qualitative data were analyzed using Open Code software (version 4.03). Audio-recorded data was directly translated from the digital recorder into English. The translated word document was converted into plain text format. The plain text format of the translated content was transported into Open Code and then coded, categorized into sub-thematic areas, and analyzed by predetermined themes using a thematic analysis approach. A triangulation protocol was used and some quotes from the qualitative data that best explain the objective were identified and presented in the participants' own words in parallel with the quantitative information to give more insight into the study.

## RESULTS

The respondents' socio-demographic characteristics are presented in Table 1. The mean age of the study subjects was 25 years ($\pm 5.07$). The vast majority (42.1%) completed primary education, while one out of five sex workers did not attend formal education. Half of the respondents were single, 40% were divorced/separated/widowed, and only 9.1% were married. More than one-third of participants had jobs other than sex work. Most (56.2%) of CFSWs had no children, while a pretty good amount (43.8%) had children. However, only 16.5% of them lived with their children/husbands, and close to half (47.8%) of them lived with peers. More than half of sex workers' estimated daily income was ≥500 Ethiopian Birr (ETB). While considering usual personal behaviors, four out of five sex workers drank alcohol regularly, one out of ten smoked cigarettes, and three out of four chewed Khat regularly. In addition to that, 30.0% of female sex workers used substances or drugs like cannabis (powder/injection), shisha, and ganja regularly. However, only 11.0% of sex workers used substances in exchange for sex, and a majority (44.1%) of sex workers didn't use any mass media (TV/Radio) or social media.

Table 2 outlines mainly sex-trade-related variables and health-hygienic behaviors of female sex workers. Most (68.7%) of CFSWs started sex work because of poverty, and a majority (81.5%) of them started it at ≥18 years. Two-thirds of the respondents had ≤4 years of experience in 'sex work'. In the last seven days prior to the survey, 81.1% of female sex workers worked for ≥3 days a week. Commercial female sex workers are mainly categorized based on their sex trade places, half of the sex workers were hotel-based, 43.8% of them were street-based sex workers, and only 6.4% worked in their homes. A majority (68.4%) of female sex workers had sex with at least two clients per day. Though their coital frequency depends on the ability of their clients to pay, 80.5% had more than one coitus per client. More than one-third of female sex workers had nonpaying clients/permanent

**Table 1 Socio-demographic characteristics and personal behaviors of the commercial female sex workers (CFSWs) in Hawassa city, Ethiopia.**

| Socio-demographic variables and personal behaviors | Frequency ($n = 297$) | Percentage (%) |
|---|---|---|
| Age (years) | | |
| <25 | 139 | 46.8 |
| 25–29 | 116 | 39.1 |
| >= 30 | 42 | 14.1 |
| Mean age (SD) = 25.0 (5.07) | | |
| Educational status | | |
| No formal education | 52 | 17.5 |
| Primary education | 125 | 42.1 |
| Secondary education | 115 | 38.7 |
| College and above | 5 | 1.7 |
| Marital status | | |
| Single | 151 | 50.8 |
| Married | 27 | 9.1 |
| Divorced/separated/ widowed | 119 | 40.0 |
| Having jobs other than sex work | | |
| Yes | 108 | 36.4 |
| No | 189 | 63.6 |
| Having children | | |
| Yes | 130 | 43.8 |
| No | 167 | 56.2 |
| Living with | | |
| Alone | 106 | 35.7 |
| Children/husband | 49 | 16.5 |
| Other CSWs | 142 | 47.8 |
| Average daily income (ETB[*]) | | |
| <500 | 126 | 42.4 |
| >= 500 | 171 | 57.6 |
| Average daily expense (ETB) | | |
| <500 | 265 | 89.2 |
| >=500 | 32 | 10.8 |
| Drink alcohol regularly | | |
| Yes | 236 | 79.5 |
| No | 61 | 20.5 |
| Currently smoke | | |
| Yes | 62 | 20.9 |
| No | 235 | 79.1 |
| Chew Khat[**] | | |
| Yes | 218 | 73.4 |
| No | 79 | 26.6 |

**Table 1** (*continued*)

| Socio-demographic variables and personal behaviors | Frequency (*n* = 297) | Percentage (%) |
|---|---|---|
| Use substances/drugs[***] | | |
|     Yes | 89 | 30.0 |
|     No | 208 | 70.0 |
| Use substance in exchange of sex | | |
|     Yes | 33 | 11.1 |
|     No | 264 | 88.9 |
| Major mass media exposure | | |
|     Nothing | 131 | 44.1 |
|     TV/Radio | 113 | 38.0 |
|     Social media (Facebook, telegram etc.) | 53 | 17.9 |

**Notes.**
[*]ETB = Ethiopian Birr.
[**]Khat = One kind of herbal plant's leave used as recreational or mood changing drug.
[***]Drugs = Cannabis (powder/injection), Shisha and Ganja.

partners. Moreover, 27.3% of sex workers in this study did not use condoms regularly or consistently.

In this study, 81.8% of female sex workers used modern contraceptive methods, while 29.3% experienced abortion in the past. Most (84.5%) of CFSWs in this study had no access to water and soap after commercial sex. However, 41.1% of respondents had a history of sexually transmitted diseases (STDs), and 16.8% had self-reported HIV-positive status. Furthermore, 17.5% of CFSWs reported the presence of known chronic illnesses like hypertension, diabetes mellitus, bronchial asthma, and cervical cancer (Table 2).

Table 3 represents the prevalence of nutritional status and minimum dietary diversity for women (MDD-W) among CFSWs. The mean (±SD) height and weight of the female sex workers were respectively 160.7 cm (±7.34) and 56.6 kg (±9.16).The mean BMI was 21.9 kg/m$^2$ (±3.1). Prevalence of underweight (CED/chronic energy deficiency) and overweight/obesity among CFSWs were respectively 14.1% and 16.8%, while a majority (69.0%) had normal BMI. Most (55.6%) of the CFSWs achieved MDD-W (*i.e.,* eaten >5 food groups yesterday), and the rest (44.6%) of them had ≤5 MDD-W. Minimum dietary diversity for women (MDD-W) of CFSWs was not associated with the prevalence of nutritional status (*P* = .097) among female sex workers.

Different food groups consumed in one 24-hour recall by CFSWs of Hawassa city are shown in Fig. 1. It was observed that 86.2% of the participants consumed starchy staples (grains/white root and tubers), 49.8% consumed pulses, and only 12.8% reported taking nuts and seeds the previous day. Regarding animal food sources, 25.3% consumed milk and milk products, 47.8% consumed meat, poultry, and fish, and 22.9% ate eggs. Similarly, four out of five sex workers consumed dark green leafy vegetables, one-third consumed vitamin A-rich fruits/vegetables, 82.5% other vegetables, and 35.7% consumed other fruits (Fig. 1).

In multivariable analyses, model-1 (normal BMI *versus* underweight) showed that living alone (Adjusted odds ratio/AOR = 0.186, (95% CI [0.043–0.804]), *P* = 0.024), chewed Khat regularly (AOR = 0.233, (95% CI [0.06–0.86]), *P* = 0.03), used substance/drugs regularly

**Table 2 Sex-trade and health-hygienic behaviors of the female sex workers in Hawassa city, Ethiopia.**

| Sex-trade and health-hygienic related behaviors | Frequency ($n = 297$) | Percentage (%) |
|---|---|---|
| Reasons for the sex work | | |
|     Lower economy/poverty | 204 | 68.7 |
|     Peer pressure | 69 | 23.2 |
|     Other reasons (harassment, exploitation) | 24 | 8.1 |
| Years of experience in sex work | | |
|     <4 years | 192 | 64.6 |
|     >= 4 years | 105 | 35.4 |
| Age entering in to sex work | | |
|     >= 18 years | 242 | 81.5 |
|     <18 years | 55 | 18.5 |
| Number days worked in the last 7 days | | |
|     <3days a week | 56 | 18.9 |
|     >= 3 days a week | 241 | 81.1 |
| Category of sex workers (usual place of sex work) | | |
|     Hotel based | 148 | 49.8 |
|     Home based | 19 | 6.4 |
|     Street based | 130 | 43.8 |
| Client per day | | |
|     One client | 94 | 31.6 |
|     >= Two clients per day | 203 | 68.4 |
| Coital frequency per clients | | |
|     One coitus per client | 58 | 19.5 |
|     >= 2 coitus per client | 239 | 80.5 |
| Non-paying partners (client) | | |
|     Yes | 102 | 34.3 |
|     No | 195 | 65.7 |
| Regular & consistent condom use | | |
|     Yes | 216 | 72.7 |
|     No | 81 | 27.3 |
| Use of modern contraceptive | | |
|     Yes | 243 | 81.8 |
|     No | 54 | 18.2 |
| History of abortion | | |
|     Yes | 87 | 29.3 |
|     No | 210 | 70.7 |
| Access to water and soap after sex | | |
|     Yes | 251 | 84.5 |
|     No | 46 | 15.5 |
| History of STI | | |
|     Yes | 122 | 41.1 |
|     No | 175 | 58.9 |

**Table 2** (*continued*)

| Sex-trade and health-hygienic related behaviors | Frequency ($n = 297$) | Percentage (%) |
|---|---|---|
| HIV status | | |
| Positive | 50 | 16.8 |
| Negative | 247 | 83.2 |
| Presence of known chronic illness* | | |
| Yes | 52 | 17.5 |
| No | 245 | 82.5 |

**Notes.**
*Chronic illness were hypertension, Diabetes Mellitus, Bronchial Asthma and cervical cancer.

**Table 3** Prevalence of nutritional status and dietary diversity score according to BMI (Kg/m²) among the female sex workers in Hawassa city, Ethiopia.

| Variables | Mean ± SD | BMI (Kg/m²) categories ($n = 297$) | | | P-value |
|---|---|---|---|---|---|
| | | Underweight n (%) | Normal n (%) | [a]Overweight/obese n (%) | |
| Height (cm) | 160.7 ±7.3 | | | | |
| Weight (Kg) | 56.6 ± 9.2 | | | | |
| BMI (Kg/m²) | 21.9 ± 3.1 | 42 (14.2) | 205 (69.0) | 50 (16.8) | |
| (Minimum-Maximum) | (14.5–32.3) | | | | |
| Mean DDS-W* [n (%)] | 4.75 ±1.5 | | | | |
| <5 DDS-W [132 (44.4%)] | | 24 (57.1) | 83 (40.5) | 25 (50.0) | [#]P=.097 |
| ≥ 5 DDS-W [165 (55.6%)] | | 18 (42.9) | 122 (59.5) | 25 (50.0) | |
| Total [297 (100%)] | | 42 (14.1) | 205 (69.0) | 50 (16.8) | |

**Notes.**
[a]Overweight (25.1–29.9 BMI) were ($n = 47$), and Obese (≥30.0 BMI) were only ($n = 03$).
DDS-W*, Dietary diversity score for women.
[#]Chi-square Test.

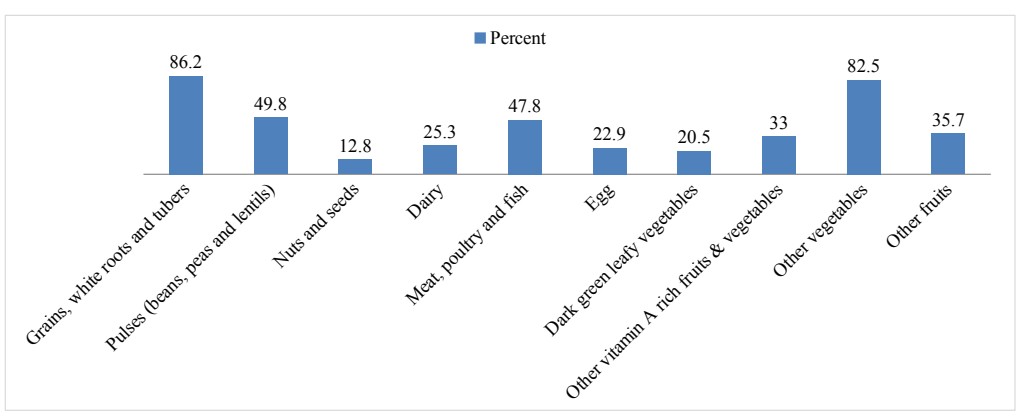

**Figure 1** Food groups consumed from 24-hour recall by CFSWs in Hawassa city, Ethiopia.

(AOR = 10.57, (95% CI [2.58–43.24]), $P = 0.001$), used drugs in exchange of sex (AOR = 4.971, (95% CI [1.279–19.322]), $P = 0.021$), and HIV-positive status (AOR = 21.64, (95% CI [5.86–79.95]), $P < 0.001$) were five significant predictors which influenced CFSWs to be
underweight (Table 4). Similarly, the normal *versus* overweight/obesity model (model-2) postulated that having jobs other than sex work (AOR = 0.108, (95% CI [0.03–0.35]), *P* = 0.000), average daily income <500 ETB (AOR = 3.02, (95% CI [1.20–7.58]), *P* = 0.018), being hotel/home-based sex worker (AOR = 12.35, (95% CI [3.83–39.7]), *P* = 0.000), and presence of any other chronic illness (*e.g.*, hypertension, diabetes) (AOR = 5.15, (95% CI [1.80–14.7]), *P* = 0.002) were significantly associated with overweight/obesity among CFSWs (Table 4).

Table 5 mainly represents the model fitting information, goodness of fit, pseudo-$R^2$, and the classification table for multinomial logistic regression (MNLR), generated from the multivariable analyses (Table 4). Here, lower AIC value in relation to a smaller log likelihood ratio/$L^2$ suggests a better model fit (AIC = 324.1 which was lower than intercept-only model/null-model having no predictors (483.7), $L^2$ = 250.5 lower than null-model (480.7)). Likelihood ratio (LR) tests showed the model containing the full set of predictors represents a significant improvement in fit relative to a null model (LR $\chi^2(32)$ = 230.210, *p* < .001). Furthermore, *P*-value of Pearson's goodness of fit is one (sig = 1.0) means could not rejecting the 'null hypothesis', which is in other word "the model fits well". Also, Nagelkerke-$R^2$ = 0.631 means 63.1% (>60.0%) of variation in the dependable variables (BMI) are explained by the independable variables. Classification table gives an indication of how well the model is performing in correctly predicting category membership on the dependable variable/DV (BMI kg/m$^2$). The overall classification accuracy for the BMI (kg/m$^2$) model was 78.5%. Clearly the model performed well in predicting who would fall into the 'normal BMI' category, followed by 57.1% underweight (CED) and 48.0% obese/overweight (Table 5).

## DISCUSSION

This study investigated the nutritional status and associated factors among commercial female sex workers (CFSWs) in Hawassa city of south Ethiopia. The prevalence of underweight and overweight/obesity among CFSWs in Hawassa city was 14.1% and 16.8%, respectively. Interestingly, self-reported HIV positivity was 16.8% (*n* = 50) among them (*n* = 297), which is reported to be lower than the previous assessment (*Ethiopia Public Health Association/EPHA, 2013*). Despite having higher HIV rates among the general population, women and sex workers in African countries (*Ethiopia Public Health Institute, 2022*; *Hladik et al., 2017*; *United Nations Programme on HIV/AIDS, UNAIDS, 2021*; *Ethiopia Public Health Association/EPHA, 2013*; *World Health Organization, 2018*) and proven synergistic relationship with nutrition, immunity, and HIV-infection (other infections/STIs) (*Nigusso & Mavhandu-Mudzusi, 2021*; *Zambia Ministry of Health, 2017*), no study is available to focus on the nutritional status and its associated factors among commercial female sex workers (CFSWs) in African regions. However, the prevalence of nutritional status among CFSWs in Asia (underweight = 22.8%) is currently available where no HIV was detected among them (*Kawser et al., 2020*). Surprisingly, the underweight rate among CFSWs of Hawassa city is slightly lower than that of reproductive-aged women in Ethiopia (*Kassie, Abate & Kassaw, 2020a*; *Kassie et al., 2020b*) and commercial female sex workers in Dhaka city, Bangladesh (*Kawser et al., 2020*). However, overweight/obesity prevalence among

**Table 4  Factors associated with the nutritional status among commercial female sex workers in Hawassa city, Ethiopia.**

| Explanatory variables | [a]Model-1 (Underweight verses Normal BMI) | | | | [a]Model-2 (Overweight/obesity verses Normal BMI) | | | |
|---|---|---|---|---|---|---|---|---|
| | B | Odds ratio | 95% CI Lower-Upper | | p-value | B | Odds ratio | 95% CI Lower-Upper | | p-value |

| Explanatory variables | B | Odds ratio | Lower | Upper | p-value | B | Odds ratio | Lower | Upper | p-value |
|---|---|---|---|---|---|---|---|---|---|---|
| Intercept | −1.281 | | | | | −3.637 | | | | |
| Age (years) | | | | | | | | | | |
| 19–24 | −.722 | .486 | .072 | 3.265 | .458 | −.806 | .447 | .091 | 2.191 | .321 |
| 25–29 | −1.526 | .217 | .035 | 1.351 | .102 | 1.108 | 3.027 | .845 | 10.843 | .089 |
| >= 30 | | | | | | | | | | |
| Having other work | | | | | | | | | | |
| Yes | .181 | .835 | .283 | 2.460 | .743 | −2.224 | .108*** | .033 | .358 | .000 |
| No | | | | | | | | | | |
| Average daily income | | | | | | | | | | |
| <500 ETB | .802 | 2.231 | .735 | 6.769 | .157 | 1.106 | 3.023* | 1.205 | 7.586 | .018 |
| >= 500 ETB | | | | | | | | | | |
| Years of experience | | | | | | | | | | |
| <4 years | .291 | 1.338 | .344 | 5.201 | .674 | −.527 | .591 | .217 | 1.610 | .030 |
| >= 4 years | | | | | | | | | | |
| Living with | | | | | | | | | | |
| Alone | −1.683 | .186* | .043 | .804 | .024 | .220 | 1.247 | .470 | 3.304 | .658 |
| Children/husband | −1.194 | .303 | .064 | 1.429 | .131 | .474 | 1.607 | .448 | 5.770 | .467 |
| Other CFSWs | | | | | | | | | | |
| Drink alcohol regularly | | | | | | | | | | |
| Yes | −.539 | .584 | .142 | 2.405 | .456 | −.244 | .784 | .263 | 2.332 | .661 |
| No | | | | | | | | | | |
| Chew Khat regularly | | | | | | | | | | |
| Yes | −1.455 | .233* | .063 | .866 | .030 | −.423 | .655 | .222 | 1.935 | .444 |
| No | | | | | | | | | | |
| Use substances/drugs regularly | | | | | | | | | | |
| Yes | 2.359 | 10.577* | 2.587 | 43.243 | .001 | −.856 | .425 | .123 | 1.468 | .176 |
| No | | | | | | | | | | |
| Use drugs in exchange of sex | | | | | | | | | | |
| Yes | 1.604 | 4.971* | 1.279 | 19.322 | .021 | .880 | 2.241 | .499 | 11.639 | .274 |
| No | | | | | | | | | | |
| Use mobile for catching clients | | | | | | | | | | |
| Yes | .669 | 1.953 | .557 | 6.851 | .296 | .564 | 1.758 | .609 | 5.077 | .297 |
| No | | | | | | | | | | |
| Usual place of sex | | | | | | | | | | |
| Hotel/home based | −1.096 | .334 | .110 | 1.013 | .053 | 2.514 | 12.357*** | 3.839 | 39.777 | .000 |
| Street based | | | | | | | | | | |

**Table 4** (*continued*)

| Explanatory variables | [a]Model-1 (Underweight verses Normal BMI) | | | | | [a]Model-2 (Overweight/obesity verses Normal BMI) | | | | |
|---|---|---|---|---|---|---|---|---|---|---|
| | B | Odds ratio | 95% CI | | | B | Odds ratio | 95% CI | | |
| | | | Lower-Upper | | p-value | | | Lower-Upper | | p-value |
| Use any modern contraceptives | | | | | | | | | | |
| Yes | −.669 | .512 | .155 | 1.691 | .272 | −.142 | .868 | .217 | 3.469 | .841 |
| No | | | | | | | | | | |
| HIV status | | | | | | | | | | |
| Positive | 3.075 | 21.649[***] | 5.862 | 79.950 | .000 | .139 | 1.149 | .347 | 3.809 | .820 |
| Negative | | | | | | | | | | |
| Presence of any Chronic illness | | | | | | | | | | |
| Yes | 1.000 | 2.719 | .601 | 12.300 | .194 | 1.640 | 5.157[*] | 1.800 | 14.775 | .002 |
| No | | | | | | | | | | |

**Notes.**

[a]Variables significant (*P* < 0.05) in bivariate analysis (Table S2) were incorporated in the Multinomial logistic regression.

95% CI, 95% Confidence Interval; ETB, Ethiopian Birr.

Chronic illness were hypertension, Diabetes Mellitus, Bronchial Asthma and cervical cancer; All the last categories of the explanatory variables were "Reference categories".

[*]*p* < 0.05.
[**]*p* < 0.01.
[***]*p* < 0.001.

CFSWs in Bangladesh (*Kawser et al., 2020*) was consistent with the current study. Another study (*Huq et al., 2013*) conducted in Dhaka showed chronic energy deficiency (CED) was much higher (57%) than this study; this might be due to the small sample size and inclusion of the drug addict groups in addition to sex workers.

Multivariable analyses outlined various factors associated with the nutritional status (Table 4) among commercial female sex workers (CFSWs) of Hawassa city. It was revealed that living with other CFSWs caused underweight, and those who lived alone had 81.5% (AOR 0.175) less probability of being underweight. It is a new finding as 'living with/without peers' was not associated with the nutritional status of CFSWs reported in the previous study conducted in a South-Asian country (*Kawser et al., 2020*); This might be due to those sex workers who lived with other CFSWs in Hawassa city sharing everything in common including meals which usually eat together outside the home, in hotels/restaurants. For most of the time, they shared food with their peers. An in-depth interview also supported this finding; a 19-year-old sex worker said, "*We share foods because it is difficult to afford separately. We share costs to buy food staffs like Injera, onion, and others*".

Notably, in this study, better nutritional status was observed among regular Khat-chewing sex workers (76.0% lower probability to be underweight) as compared to those who did not chew 'Khat '. 'Khat leave' is known as a recreational drug in East African countries including Ethiopia and most of the people of these countries chew 'Khat' to elevate moods; this may be due to Khat chewing sex workers who might sit for long periods in Khat chewing places, thus spending more 'sedentary/lazy' time, which might result in weight gain. In-depth interview, a 35-years-old sex worker in Adare General Hospital said about Khat, "*When we get money, we go out and eat and drink together with our friends. Then, we come back to our rooms and chew Khat until evening. That is all*". The present study also observed that female sex workers who used 'substances/drugs' and 'took

**Table 5  Model fitting information, goodness of fit, pseudo-R², and the classification table for multinomial logistic regression (MNLR) (extracted from the analyses of Table 4).**

| Model fitting criteria | | | | Likelihood ratio tests | | |
|---|---|---|---|---|---|---|
| | AIC | BIC | −2 Log Likelihood | | | |
| Intercept Only | 483.356 | 490.356 | 480.742 | Chi-square | def | sig |
| Final | 324.114 | 449.701 | 250.532 | 230.210 | 32 | .000 |
| **Goodness of fit** | | | | Chi-square | def | sig |
| Pearson | | | | 141.259 | 220 | 1.000 |
| Deviance | | | | 129.276 | 220 | 1.000 |
| **Pseudo R-square** | | | | R² value | | |
| Cox and Snell | | | | 0.511 | | |
| Nagelkerke | | | | 0.631 | | |
| McFadden | | | | 0.430 | | |
| **Classification table for dependent variable BMI (kg/m²)** | | | | | | |
| Underweight/chronic energy deficiency (CED) | | | | 57.1% | | |
| Normal BMI | | | | 90.2% | | |
| Overweight/Obese | | | | 48.0% | | |
| Overall Percentage | | | | 78.5% | | |

**Notes.**
def, Degree of freedom; sig, significant; AIC, Akaike Information Criterion; BIC, Bayesian Information Criterion.
Lower AIC value suggests a better model fit (AIC = 324.1 lower than intercept = 483.3).
*P*-value of Pearson's goodness of fit is one (sig = 1.0) which means non-significant, they provide further evidence of a well-fitting model.

substances/drugs in exchange for sex' were respectively 11.7 and 4.2 times more likely to be associated with being underweight. A higher prevalence (62%) of malnutrition was also observed among drug addicts (*Huq et al., 2013*) and could indicate the possible association with malnutrition. The study conducted among CFSWs in Dhaka city also reported that those addicted to cannabis were more likely to be associated with being underweight and significantly decreased the probability of being overweight/obese (*Kawser et al., 2020*). A recent narrative review on 'nutritional status and eating habits of drug addicts' revealed that the use of chronic substances affects a person's nutritional status and body composition through decreasing intake, nutrient absorption, and deregulation of hormones that alter the mechanisms of satiety and food intake (*Mahboub et al., 2020*). More importantly, female sex workers paid a small portion of money for food (than khat/drugs) from their daily income; they also reduced the frequency and amount of their meals depending on catching clients. According to the key informant interviews of this study, a service provider in Millennium health center said, *"I realize that it would be better for their health if they expend more money for food than Khat and drugs. Most of them expend their money for their addictions"*.

This study showed that self-reported HIV status was strongly associated with being underweight ($P = 0.000$) among CFSWs. Sex workers with positive HIV status had a 21.6-times higher odds ratio to being underweight than HIV-negative CFSWs, which agrees with recent studies (*Hussien et al., 2021*; *Nigusso & Mavhandu-Mudzusi, 2021*). A higher prevalence of malnutrition among people living with HIV and AIDS (PLWHA) is reported

in Ethiopia (*Hussien et al., 2021*; *Nigusso & Mavhandu-Mudzusi, 2021*), which also observed in this study as a strong predictor of undernutrition among HIV-positive sex workers. This finding is also supported by the qualitative part of this study, which found that 'sexually transmitted infections/STIs', and HIV are the most common illness among malnourished CFSWs. A peer educator at Adare general hospital said, "*Sexually transmitted infections and HIV are the most common infections observed among sex workers*". However, most HIV-positive sex workers hardly adhere to ART treatment. A 30-year-old sex worker said, "*I was tested HIV positive six years ago and gave birth to an HIV-negative child. There are also many HIV-positive sex workers. Furthermore, some sex workers on ART medical treatment put their drug in the home in secret*". A sex worker at Millennium health center also said, "*HIV-positive individuals, in particular sex workers, do face shortage of food while receiving their drugs. Though they take* medicine *daily, they have several addictions and food shortages. Because of this, they throw away* their *medicine*".

Overweight/obesity prevalence among sex workers seems to get less attention than bundles of problems around them. The present study evidenced that sex workers having jobs (*e.g.*, waitresses/cleaners in hotels and restaurants/pensions, broker agents, and small businesses) other than sex work had 89% (AOR 0.11) less likelihood of being overweight/obese as compared to jobless CFSWs; this may be related to 'physical exercise/activities', indicating having jobs other than 'sex work' might have an 'equivalent effect of exercise' on the nutritional status of CFSWs, and in agreement with a recent study of Nepal (*Tharu & Mahatra, 2021*). Similarly, lower daily income ($\leq$500 ETB) and being hotel-based/home-based sex workers were significantly (Adjusted odds ratio/AOR 3.0 and 12.3, respectively) associated with overweight/obesity. The study reported that hotel-based sex workers had higher income and social status and were likelier to be overweight/obese than on-street CFSWs (*Kawser et al., 2020*); this might be due to hotel-based sex workers having a more stable work environment compared to street-based sex workers. Key informant interviews also support this; the peer educator in Adare General Hospital also said, "*By the way, there is a status difference among hotel-based and street-based sex workers. Hotel-based sex workers may get income in thousands of Birr, and home-based sex workers might get paid double, including their room fee, but street-based sex workers get paid the least*". Sex workers with the presence of known chronic non-communicable diseases/NCDs (*e.g.*, hypertension, diabetes) had a five times higher odds ratio to be overweight/obese as compared to having no chronic illness. This finding is consistent with several studies in Ethiopia (*Tesfa & Demeke, 2021*; *Abdissa, Dukessa & Babusha, 2021*). An Ethiopian meta-analysis showed a statistically significant association between BMI and hypertension (*Tesfa & Demeke, 2021*). Another study in southwest Ethiopia revealed that being comorbid, having higher income, and having a family history of overweight and obesity were significantly associated with BMI (*Abdissa, Dukessa & Babusha, 2021*). The qualitative part of the present study also echoed this and identified that no 'nutrition counseling/education' was provided to female sex workers in all three study clinics. The training packages for 'key population service provider' has several components. Unfortunately, nutrition counseling/education and nutrition intervention were not incorporated in the 'training and service packages' provided for key populations.

Dietary intake is an essential factor in determining nutritional status. In the current study, the mean individual dietary diversity score for women was not up to the mark (4.75 ±1.5) for sex workers, despite most (55.6%) of CFSWs achieving minimum dietary diversity for women/MDD-W (consumption of ≥5 food groups (*Food and Agriculture Organization, FAO, 2016*)). However, it was noticed that MDD-W was not associated with the prevalence of nutritional status (*P* = .097) among CFSWs of Hawassa city. This may be because all the sex workers of Hawassa city, Ethiopia, use more non-food items like alcohol, Khat, or drugs/substances daily than usual food items. As mentioned before, it is part of their lives and is also supported by the qualitative part of this study. The Millennium Key Population clinic health officer said, *"Sex workers usually wake up from sleep late in the morning, and then, they eat breakfast"*. After breakfast, they chew Khat and smoke shisha in a group. Some of them may smoke cigarettes. Then, they start drinking alcohol early in the evening". Dietary diversity especially for sex workers had not been assessed before in Ethiopia, even though in greater Africa. Studies conducted in Africa (*Anyanwu et al., 2021*; *Gitagia et al., 2019*) reported that the proportion of minimum DD-W was low (<5 food groups) and not meeting the minimum DD threshold for women (≥5 food groups). Most (75%) Kenyan women of reproductive age (*Gitagia et al., 2019*) and 12.9% to 20.3% of pregnant and lactating women of Ethiopia (*Anyanwu et al., 2021*) consumed <5 food groups daily.

## CONCLUSIONS

Commercial female sex workers (CFSWs) in this study faced a double burden of malnutrition. The prevalence of underweight and overweight/obesity among CFSWs in Hawassa city was 14.1% and 16.8%, respectively. Varieties of socioeconomic, lifestyle and health-related factors influenced the nutritional status among CFSWs of Hawassa city. Living alone, chewing 'Khat' regularly, regular use of drugs, use of drugs in exchange for sex, and positive HIV status were associated with being underweight/CED. In addition, not having other jobs, higher daily income, being a hotel/home-based sex worker, and chronic illness was associated with being overweight/obese. An in-depth interview of the CFSWs postulated that 'lack of money and food' was the leading cause of starting 'sex work'. Hotel/home-based sex workers had better income, social status, and safety than street-based sex workers. In their daily routine of life, alcohol, 'Khat', and substances/drugs are used more frequently than foods. HIV is the most common illness among sex workers. In addition, this study also explored the lack of nutrition counseling, services, and interventions for commercial female sex workers at health facilities.

Policy reforms to remove punitive approaches to sex work, ensure supportive policies, standard workplace, and foster sex workers' (SW) ability to work collectively are recommended (*Goldenberg, Duff & Krusi, 2015*). Sex workers need accessible, acceptable, and good-quality medical care at all levels, which may be provided through a variety of channels such as sex worker-led services, including clinical services, harm reduction, and drug treatment services, and integrated services at sexual, reproductive, and primary healthcare centers (*World Health Organization, WHO)(2012*).

## ACKNOWLEDGEMENTS

First of all, our special thanks go to almighty God. Also, thanks to the School of Nutrition, Food Science and Technology of Hawassa University for administrative support. We would also like to thank staff members of Adare General Hospital, Millennium Health Center, and Family Guidance Association of Ethiopia, Southern Area Office, for their support during data collection. Finally, we would like to thank the participants of this study who were willing to cooperate during data collection.

### Funding

The authors received no funding for this work.

### Competing Interests

The authors declare there are no competing interests.

### Author Contributions

- Zegeye Gelan conceived and designed the experiments, performed the experiments, analyzed the data, prepared figures and/or tables, authored or reviewed drafts of the article, and approved the final draft.
- Zelalem Tafese conceived and designed the experiments, performed the experiments, analyzed the data, prepared figures and/or tables, authored or reviewed drafts of the article, and approved the final draft.
- Eskinder Yilma performed the experiments, analyzed the data, authored or reviewed drafts of the article, and approved the final draft.
- Mahbuba Kawser analyzed the data, authored or reviewed drafts of the article, and approved the final draft.

### Human Ethics

The following information was supplied relating to ethical approvals (i.e., approving body and any reference numbers):

Ethical clearance to conduct the study was obtained from the Hawassa University, School of Graduate studies (ref No: IRB/021/14).

### Ethics

The following information was supplied relating to ethical approvals (i.e., approving body and any reference numbers):

Ethical clearance to conduct the study was obtained from the Hawassa University, School of Graduate studies (ref No: IRB/021/14).

### Data Availability

The raw data are available in the Supplemental Files.

## Supplemental Information

Supplemental information for this article can be found online at http://dx.doi.org/10.7717/peerj.15237#supplemental-information.

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
