# Peer review of "Nutritional status and its associated factors among commercial female sex workers in Hawassa city, south Ethiopia"

_PeerJ, doi:10.7717/peerj.15237_

## Round 0.1 · original submission · Major Revisions

I read your manuscript and the review reports from two expert reviewers. I also think this manuscript includes interesting data. Both reviewers are asking you for significant changes to the manuscript. I agree with both reviewers' opinions. I request the current revisions as noted by the reviewers.

Reviewer 1 ·

Basic reporting

1. The manuscript is well-written but also comes along with minor grammatical errors in some sentences.
2. The background information is written in detail.
3. Acronyms that appear for the first time should come along with the full description.
4. The experiment results should have a better layout that is easier for readers, concise figures and tables are highly recommended to demonstrate the result
5. All the literature references in this paper lack numbering which causes difficulties in tracking.

Experimental design

1. The primary research aim and data collection are clearly defined and described.
2. The statistical analysis part needs more details, for instance, in section 2.6 multiple statistical methods Kolmogorov-Smirnov Goodness-of-Fit Test, chi-square test and multinomial logistic regression are mentioned and used. It would be great if the author is able to talk a little bit about this method and why this method.

Validity of the findings

1. Although the authors made enough effort in the background introduction and storytelling, the impact on the research is limited. As the authors mentioned in the paper, sex workers basically have financial difficulties so it’s not very surprising they are malnutrition.
2. Again, the statistical method detail is insufficient, the methodology should be clarified as well.
3. Table 1 - table 5 only demonstrates sex workers’ personal living habits and corresponding statistical analysis, figure 1 shows the percentage among the different dietary groups. My concern is despite other factors, diet plays an important role in nutrition status, so it’s worth including an analysis here.

·

Basic reporting

The topic of the article is interesting but I had difficulties reading it because of the style of writing. The introduction contained too many references and was in my view not a good overview of the main discussions in the field. In addition, the English was not always perfect which affected the readability of the paper. Last but not least the paper is mainly based on quantitative research and I am not able to assess the validity of it. I am a qualitative researcher and it was very hard for me to read and understand the statistical analysis. The authors do give sufficient background and context although I am not sure it will be clear for people that do not know the Ethiopian context.

Experimental design

Unfortunately I am unable to assess this because the paper is mainly of a quantitative nature. Yet, I do think the problem is well defined but the way in which the research has been done and the validity of the findings cannot be assessed by me.

Validity of the findings

No comment because I am unable to assess the findings

---

## Round 0.2 · Minor Revisions

The manuscript has undergone significant improvement. However, there are still some places that need revision before a final decision can be made. Please consider the feedback provided by Reviewer 1 when revising the manuscript.

Reviewer 1 ·

Basic reporting

1. The manuscript is well-written and the structure of this manuscript is better than the previous version, the background is more apparent.

Experimental design

1. The data is well collected and described in detail.
2. The statistical analysis section is clearer and the results are well discussed

Validity of the findings

1. The model-building and selection procedure should be explained. Also, the model interpretation should be included

---

## Round 0.3 · accepted · Accept

Congratulations!

I have reviewed all the comments from you and the reviewers, and have deemed ti acceptable for publication in PeerJ.

Thank you for your submission.

Reviewer 1 ·

Basic reporting

1. The manuscript is well-written and easy to understand. The background information is written in detail.

Experimental design

1. The modifications of the statistical analysis part based on previous review feedback are clear.

Validity of the findings

No comment